# Could Photodynamic Therapy Be a Promising Therapeutic Modality in Hepatocellular Carcinoma Patients? A Critical Review of Experimental and Clinical Studies

**DOI:** 10.3390/cancers13205176

**Published:** 2021-10-15

**Authors:** Abhishek Kumar, Olivier Moralès, Serge Mordon, Nadira Delhem, Emmanuel Boleslawski

**Affiliations:** 1INSERM, CHU-Lille, U1189-ONCO-THAI–Assisted Laser Therapy and Immunotherapy for Oncology, University of Lille, F-59000 Lille, France; abhishek.kumar@ibl.cnrs.fr (A.K.); olivier.morales@ibl.cnrs.fr (O.M.); serge.mordon@inserm.fr (S.M.); 2CNRS UMS 3702, Lille Institute of Biology, F-59021 Lille, France; 3INSERM U1026 Bioingénierie Tissulaire (BioTis), Université de Bordeaux, F-33076 Bordeaux, France; 4Department of Digestive Surgery and Liver Transplantation, Hôpital Huriez, Nord-de-France University Hospital, CHU Lille, Service de Chirurgie Digestive et Transplantation, F-59000 Lille, France

**Keywords:** anti-cancer therapy, anti-tumor immunity, cirrhosis, active targeting, passive targeting

## Abstract

**Simple Summary:**

Hepatocellular Carcinoma (HCC) is one of the leading causes of cancer-associated mortality worldwide. With a limited number of therapeutic options available and a lack of effective anti-tumoral immune responses by the therapies, there is a dire need to search for new translational treatment options. Photodynamic Therapy (PDT), in recent years, has proven itself as an effective anti-cancer therapy. In this review, we discuss the mechanism of PDT, its evolution as an anti-cancer modality, with a special focus on HCC. We also highlight the immune response generated by PDT and how it could be essential in HCC treatment. Finally, we proposed an intraoperative procedure for the treatment of HCC by combining hepatectomy with PDT.

**Abstract:**

Photodynamic Therapy (PDT) relies on local or systemic administration of a light-sensitive dye, called photosensitizer, to accumulate into the target site followed by excitation with light of appropriate wavelength and fluence. This photo-activated molecule reacts with the intracellular oxygen to induce selective cytotoxicity of targeted cells by the generation of reactive oxygen species. Hepatocellular carcinoma (HCC), one of the leading causes of cancer-associated mortality worldwide, has insufficient treatment options available. In this review, we discuss the mechanism and merits of PDT along with its recent developments as an anti-cancerous therapy. We also highlight the application of this novel therapy for diagnosis, visualization, and treatment of HCC. We examine the underlying challenges, some pre-clinical and clinical studies, and possibilities of future studies associated with PDT. Finally, we discuss the mechanism of an active immune response by PDT and thereafter explored the role of PDT in the generation of anti-tumor immune response in the context of HCC, with an emphasis on checkpoint inhibitor-based immunotherapy. The objective of this review is to propose PDT as a plausible adjuvant to existing therapies for HCC, highlighting a feasible combinatorial approach for HCC treatment.

## 1. Introduction

For hepatocellular carcinoma (HCC) treatment, the Barcelona Clinic Liver Cancer (BCLC) staging system is the most commonly used staging classification, which classifies the patients into Early, Intermediate, Advanced, and End-Stage [1,2,3,4]. In a large international prospective cohort of patients with HCC, the proportions of patients with BCLC stages A, B, C, and D at the time of diagnosis were 19.9%, 1.8%, 75.6%, and 2.6%, respectively [5]. Based on this staging system, curative options are only considered in early-stage HCC. Patients with solitary HCC and preserved liver function are referred to either surgical resection or percutaneous ablation while liver transplantation, when feasible, is reserved for up to three nodules of no more than 3 cm in diameter. In every other situation, which concerns the majority of patients with HCC, only palliative options are available, ranging from locoregional therapies such as chemoembolization to targeted therapies such as *sorafenib*, if not best care support. The median survival gain obtained with such treatments, although statistically significant, does not exceed one year for patients with advanced tumors [6,7,8]. Furthermore, even in the early stages and after treatment with curative intent, HCC recurrence rates are quite high and re-treatment is not always feasible. Patients treated with hepatectomy experience a tumor recurrence rate of about 70% at 5-years [9]. However, in the therapeutic armamentarium used in managing HCC, other emerging therapies have shown encouraging results. Those based on tumor radiation, i.e., stereotaxic body radiation therapy [10] and trans-arterial radioembolization with ^90^Ytrium [11] are still under clinical evaluation, and it is unknown whether radiation therapies would outperform surgery or thermal ablation in the future. In addition, those treatments have specific contraindications. Therefore, there is still a need to explore or develop novel therapeutic modalities for HCC, either alone or combined with locoregional treatments.

Photodynamic Therapy (PDT) is a clinically approved anti-cancer treatment for certain neoplasms, such as advanced cancer of the esophagus and certain early- and late-stage lung tumors. It relies on the systemic or topical administration of a non-toxic dye called photosensitizer (PS) which accumulates in the target site during a predetermined duration, called the drug-to-light interval. At the end of this period, the target site is illuminated by the light of appropriate wavelength and energy corresponding to the PS resulting in PS photo-excitation [12]. This excited PS shall further transfer its energy to surrounding intracellular oxygen, which thereby forms reactive oxygen species (ROS) such as peroxide, singlet oxygen, and hydroxyl species, to finally induce a cytotoxic effect. These PSs must exhibit high and selective accumulation in the tumor along with low or minimal dark toxicity (i.e., the toxicity induced by the PS in the absence of illumination), high bio-stability, and high bio-clearance [12,13]. PDT can either directly induce cell death by necrosis or apoptosis or both, or indirectly by targeting the tumor microenvironment and vasculature to induce an inflammatory and immune response against the tumor [13,14]. It should be noted that to inflict vascular damage, there should be a very short drug-to-light interval (0–30 min) with a PS of fast bio-clearance. Upon photo-excitation, the PS, which is still circulating in the vascular compartment, shall cause vascular damages through low-density lipoprotein receptor-mediated endocytosis pathways and lead to thrombosis and microvessel occlusion [15].

In this review, we will unveil the role of PDT as an anti-cancer therapeutic modality and describe how the main obstacles against its development have been or could be countered. We will then review the use of PS and PDT for HCC diagnosis and treatment, and discuss possible future research endeavors in this field, including the impact of PDT for inducing an anti-tumor immune response.

## 2. Development of Photodynamic Therapy as an Anti-Cancer Treatment

PDT was first coined by Hermann von Tappeiner in 1903 [16] and was initially used to treat cutaneous disorders. PDT soon gained widespread usage in the field of dermatology to treat various skin disorders namely, papillomavirus infections, cutaneous leishmaniasis, actinic keratoses, acne, viral warts, photo-rejuvenation, psoriasis, hypertrophic, keloid scars, and port wine stains, with guidelines and recommendations being developed by agencies of different countries [17]. PDT in ophthalmology has been employed since the 1990s as a treatment option for Choroidal Neovascularization (CNV), Age-related Macular Degeneration (AMD) and recently being investigated for Polypoidal Choroidal Vasculopathy (PCV) [18]. Some of the classical PSs used here are pro-drug 5-Aminolevulinic Acid (5-ALA) and its hydrolyzed methyl ester Methyl Aminolevulinate (MAL), along with Photofrin^®^, chlorin, phthalocyanine, Verteporfin (Visudyne^®^), and Hematoporphyrin Mono-Methyl Ether (HMME) [17].

A major breakthrough to introduce PDT as a potential anti-cancer therapy came in 1978 when T.J. Dougherty et al. used hematoporphyrin derivative-based PDT for treatment of cutaneous or subcutaneous tumors including breast, colon, and prostate metastases [19]. This hence led to the clinical development of PDT for melanomas and squamous cell carcinomas. Since the pioneering work of Dougherty et al., PDT is now proposed as an alternative tool in cancer treatment with the introduction of laser and optic fiber-based light delivery systems and the discovery of new PS. The past decade has observed a huge number of PS being developed and tested which include nanoparticles and chemically conjugated PS. Some of them have been approved by the regulatory authorities of various countries for clinical studies, leading to a surge in the number of publications for PDT unraveling its various aspects ranging from its mechanisms of action to the possible activation of an anti-tumor immune response. Interestingly, upon illumination, almost all the PSs are degraded by photobleaching which adds to an important aspect of drug bio-clearance [20,21]. Besides the usual use as an anti-cancerous therapeutic agent, the fluorescence of the PS can be utilized either as a diagnostic agent or as an aid during surgery as it could delimit tumor burden [14,21].

The effectiveness of the therapy depends on the accumulation of the PS into the neoplasm, adequate uniform dosage, the power of the light, its penetration into the tissues, and availability of intracellular oxygen [12,14,22]. With an improved understanding of the biology and mechanism of the therapy along with the development of targeted PSs and efficient light delivery systems, the overall efficacy of the therapy has increased by overcoming some obstacles. PS penetration for various skin disorders has been enhanced by various physical and chemical pre-treatments which include enhanced drug formulation, stratum corneum removal, iontophoresis, and temperature modulations [23]. Further, the conjugation of classical PS with monoclonal antibodies, ligands, biomolecules, liposomes, and nano-carriers to increase hydrophobicity and selective accumulation of the PS has given improved results [21]. Azaïs et al. used a new folic-acid coupled PS-based intraperitoneal PDT to specifically target epithelial ovarian cancer, which has a higher folate receptor expression. Their study highlighted higher PS accumulation than non-coupled PS, along with increased human peripheral blood mononuclear cell (PBMC) proliferation [24]. Such highly targeting PS could prove to be a real asset for ovarian cancer treatment and management since most of the patients also show microscopic peritoneal metastases, which are tough to visualize and remove by surgery. A very recent investigation from our team has suggested that PDT using folate coupled PS, is an effective therapy in the treatment of pancreatic adenocarcinoma (PDAC), also activates the immune system, and could be considered as a real adjuvant for anti-cancer vaccination. Folate binds to FOLR1, which is expressed in 100% of PDAC or over-expressed in 30% of cases. Further, they observed a significant increase in the proliferation of activated-human PBMC and T cells when cultured with conditioned media of PDAC cancer cells subjected to PS-FOL/PDT [25]. This is quite an important study, as it highlights the applicability of a targeted PDT for a solid tumor in the abdominal cavity, similar to liver cancers. For delivery of homogenous illumination, optic fiber woven-based light-emitting flexible fabrics have been shown to give higher output in terms of fluorescence rate, illumination homogeneity, and flexibility [26,27]. Additionally, fractionation of the illumination dosage further increases the effectiveness of the PDT, since it results in a continuous supply of an important modulator of PDT mediated cell death, oxygen; along with an increased influx of the PS in certain cases [28]. This accompanied with an optimal illumination dose, decreases the heat generated and the underlying pain, which is a common issue during PDT treatment of skin disorders.

The cytotoxic ROS generated by PDT not only kills the tumor cells but also damages the microvasculature of the tumor. The damages inflicted upon the vascular basement membrane cause permeability of the vessels along with vasoconstriction which ultimately leads to tumor destruction [29]. Since PDT consumes the oxygen of the tumor microenvironment, this might lead to a release of angiogenic growth factors, to favor angiogenesis and facilitate the growth of remaining tumor cells, thus reducing the efficacy of the therapy [30]. However, this impact can be reduced by combining the therapy with an anti-Vascular Endothelial Growth Factor (VEGF) targeted therapy. Such combinatorial approaches with classical therapies to improve the overall cytotoxicity of PDT has given promising results [31,32].

In clinical practice, however, PDT has remained a rescue therapy in patients presenting with advanced disease, not eligible or irresponsive to conventional treatments, or being too sick to undergo surgery [14,22]. A prime reason for this is the difficulty to establish standard treatment conditions. Given the possible variation of several parameters, including the type of PS used, light dosage, power of illumination, light fractionation, drug-to-light interval; it becomes a challenging process to standardize treatment conditions for a clinical set-up in different types of cancer. Additionally, only a few PSs have been approved for clinical trials for cancer treatment, which include Porfimer sodium, Temoporfin, 5-ALA, and MAL, which thereby limit the usage of the new third-generation of PSs being developed.

However, some studies have demonstrated the benefit of PDT in a clinical setting, not only in palliative situations but also in early stages and even as an adjuvant therapy associated with surgery. Cuenca et.al. used Photofrin^®^ PDT in a small group of patients with chest wall progression of breast cancer. They observed a significant decrease in tumor size in all patients [33]. Moole et al. pooled the outcomes of 10 different clinical studies of PDT in patients with non-resectable cholangiocarcinoma, and conclude that PDT, in combination with biliary stenting, improves biliary drainage and thereby increases patient survival [34]. Gonzalez-Carmona et al. used a combination of PDT with different porphyrin derivatives with systemic chemotherapy in patients diagnosed with extrahepatic cholangiocarcinoma and demonstrated a significantly higher median survival compared with the chemotherapy-alone group, which also had a lower survival when compared with the PDT-alone group [35]. This study gives crucial evidence for the role of PDT as a local therapy to control the progression of extrahepatic cholangiocarcinoma and increase overall survival. A Photofrin^®^ II-mediated PDT in patients with breast cancer showed that 50% of the patients were responsive to the therapy with higher efficacy in case of minimal or moderate tumor extent when compared with advanced stages [36]. In another clinical study, 5-ALA injection 5 h prior to a laparoscopic surgery of ovarian cancer patients, aided tumor visualization by fluorescence through protoporphyrin IX (PpIX) providing higher sensitivity for the detection of intraperitoneal metastases [37]. A recent clinical trial coordinated by Pr. N. Reyns in our department, which is currently ongoing (NCT03048240), aimed to analyze the impact of intraoperative 5-ALA PDT delivered in the tumor bed during resection of glioblastoma, the preliminary results of which have been recently published [38]. Inspired by the promising results of this study, our team has now started the IMPALA clinical trial, where the malignant pleural mesothelioma patients shall be orally administered with 20 mg/kg of 5-ALA (Gliolan^®^), 4 to 6 h prior to thoracoscopy. After fluorescence-guided surgical intervention, the pleural cavity shall be illuminated for PDT-mediated tumoral cytotoxicity of residual tumor. Seven to ten days after the thoracoscopy, patients shall be treated with anti-PD-1 *nivolumab* immunotherapy every two weeks for up to two years (NCT04400539). For such clinical applications, our team has developed a warp-knitted light-emitting fabric and a balloon-based trocar attached illumination device, which can be coupled with endoscopic devices for intraoperative PDT protocols [39,40].

A summary of all the above-mentioned clinical trials is reported in Table 1. Overall, PDT has a mixed success–failure story in clinics, where it shows a high success rate in dermatology and ophthalmology but conflicting results have been reported for certain solid neoplasms and therefore more clinical studies are awaited with a higher number of patients to be more conclusive.

## 3. Is Photodynamic Therapy Applicable in Patients with HCC?

Recent advances in HCC treatment rely on physical therapies, such as trans-arterial radioembolization with ^90^Ytrium, improved molecular targeted therapies such as multi-kinase inhibitors, and immune-modulation by anti-Programed Death Ligand 1 (anti-PD-L1) or anti-Cytotoxic T-Lymphocyte Antigen 4 (anti-CTLA4). In this context, PDT, as being a complex product of physics, chemistry, and biology, may provide a combined local and systemic approach to HCC treatment. However, the use of PDT for the treatment of liver tumors has been limited so far. Generally, the major issues with the first generation of PS were short wavelength of absorbance, poor in-vitro aqueous stability with short circulation half-life, lesser tumor selectivity, and skin phototoxicity [12,13]. This further aggravates when using PDT on the liver, where the high vasculature makes certain PS accumulate not only in the tumor but also in the healthy parenchyma. For instance, during 5-ALA PDT for HCC, we can observe higher accumulation in the healthy liver, since the liver is the center for heme synthesis, which is used to metabolize the pro-drug 5-ALA to the actual PS, protoporphyrin IX (PpIX) [41]. This heme biosynthesis pathway is further responsible for liver pigmentation thus altering the light penetration which decreases as the function of distance [42].

The latter issue may be addressed by using PS being more homogeneously distributed throughout the tumor at an optimal concentration and requiring a higher wavelength for their activation [41]. For instance, a study on rat liver reveals that meta-tetra(hydroxyphenyl)chlorin (mTHPC) requires less PS dosage than Photofrin^®^ and other hematoporphyrin-derived PS since it activates at a higher wavelength [41,42,43]. Moreover, this cytotoxic effect was further increased with near-infrared PS 5,10,15,20-tetrakis(m-hydroxyphenyl)bacteriochlorin (mTHPBC), a PS belonging to the same class of hydroporphyrins as mTHPC but with a higher wavelength of activation light [41,42,43].

Tumor selectivity and targeting could be facilitated by coupling PSs with nano-carriers. With the recent advances in nano-carrier technology, a lot of PSs are being modified and tested for enhanced efficacy. Wang et al. demonstrated that PDT mediated by IR780 and near-infrared (NIR) illumination could induce higher cell growth inhibition of HCC cell lines when delivered by a nanoparticle complex (Pullulan, Pluronic F68, and phospholipid) also encapsulating paclitaxel, with respect to IR780 or paclitaxel alone. In-Vivo studies further demonstrated reduced tumor growth and angiogenesis [44]. Zhang et al. further combined multiple approaches of hypoxia, PDT, and chemotherapy with an efficiently designed drug delivery system based on DNA aptamers and gold nanoparticles, to develop a targeted and effective HCC therapy [45]. A list of the various photosensitizers tested on HCC, hepatoma, or healthy liver is reported in Table 2.

Indocyanine green (ICG), a water-soluble tricarbocyanine dye, is a widely used agent in clinical practice for intraoperative HCC visualization [51] and liver function assessment [52]. Additionally, it has also been used for NIR PDT of several cancer models including HCC. Interestingly, when photoactivated, ICG also generates heat, which thereby contributes to a tumor-suppressive effect, known as PhotoThermal Therapy (PTT). Under PTT the PS is photo-excited, to generate vibrational energy in the form of heat, thereby inducing a cytotoxic effect [53]. An ICG-Lactosome nanoparticle complex has been developed showing higher accumulation in HCC, improved tumor visualization, and causing higher cell death when compared with ICG alone [46]. Therefore, ICG seems to have the potentiality to become the best candidate for PDT in HCC treatment. Nevertheless, the mixed impact of PDT and PTT has its equal drawbacks. As summed up by Giraudeau et al., ICG exhibits phototoxicity via PDT at a low power dose, and via PTT at a high dose; which are both relying on different molecular and cellular mechanisms [53]. It is still controversial to comment which effect is superior to other but the efficiency of ICG-induced phototoxicity, especially by PDT, is not particularly effective, due to various reasons. Coupling it with nano-carriers might improve the targeting by binding to specific receptors, but it requires more modifications so as to improve its ROS yield and stability at physiological conditions and also avoid agglomeration [46,53].

Besides PS modification, PS-specific drug formulations can also be developed to enhance tumoral accumulation. For instance, 5-ALA can be administered with an iron-chelating agent which can eliminate the iron in the microenvironment, thereby inhibiting the metabolism of PpIX into heme and increasing PpIX accumulation [54]. In an in-vivo study, Chang et al. highlighted the use of iron chelator, 1,2-diethyl-3-hydroxypyridin-4-one (also known as CP94), caused double PpIX-based fluorescence than 5-ALA alone group and exhibit reduced skin photosensitization [55]. Vitamin D has also been proven to enhance PpIX levels in the cells [56]. Such drug formulations have not been studied for HCC but can significantly improve the efficacy of the therapy by augmenting the tumoral PS accumulation and decreasing non-tumoral cytotoxicity.

In-Vitro studies present a major setback for PDT research, the most significant being oxygen availability, since most of the cancers, including HCC, develop in a hypoxic background. The use of hypoxic chambers and organoids based 3-D cultures might prove beneficial for HCC modeling. Such systems, however, will not be cost-effective and require high skill sets. That is why a pre-clinical setup, using various humanized mouse models can help us understand the applications of PDT by giving a more detailed effect on the 3-D microenvironment. The most widely used models for this purpose are the subcutaneous tumors developed by either injecting human or murine HCC cell lines beneath the skin or transplanting small tumor pieces from one mouse to another. Many teams have developed orthotopic mouse models where the tumors are injected into the organs of origin, which gives a better model of cancer. Another approach can be the use of specific carcinogens to induce organ-specific cancers. However, these models can have major drawbacks as the light might not penetrate to its full efficiency thereby limiting the effectiveness of the therapy [47]. Our unpublished data have revealed that when excited with blue light, the fluorescence from PpIX could not be observed from the exterior but was successfully detected after the sacrifice and recuperation of the tumor core from the mice. Studies by two independent groups using 5-ALA PDT for different pre-clinical HCC models demonstrated a fluorescence-based selective accumulation of PpIX in the tumor, along with an anti-tumor effect [47,48]. The most interesting feature of these studies was PpIX accumulation and necrosis in the tumor core (up to 8 mm for mouse model). This reflects the penetrating capabilities of 5-ALA PDT, rather than a mere superficial effect.

Hepatic resection has become a standard HCC treatment for early-stage patients, even in the presence of liver cirrhosis. However, long-term survival is often limited by intra-hepatic recurrences, which are not always prevented by anatomical resection or adequate surgical margins. Furthermore, small satellite nodules are hardly detected through visual inspection and intraoperative ultrasonography, especially in liver cirrhosis [47]. Hence, PDT can introduce itself as an intraoperative procedure to hepatic surgery, where it may be used both as a simple and rapid real-time fluorescence-based visual aid and as a complementary treatment targeting the tumor surrounding parenchyma. Fluorescence-guided hepatectomy using ICG has already become a standard, and one can take advantage of its potential as a PS to perform PDT during ICG-guided surgical resection. However, as discussed above, ICG might not be the ideal PS for intraoperative PDT. By contrast, 5-ALA PDT may be the best option with less photothermal-induced injury, especially to the biliary tract. In addition, to increase tumor selectivity, one could imagine the PS to be injected right into the tumor bed with the help of a trans-arterial catheter, a practice commonly used during trans-arterial chemo-embolization. At the end of the surgery, immediately after removal of the specimen, all conditions would be met to perform an efficient adjuvant PDT. The tumor bed would be exposed with clear access to the remaining peri-tumoral parenchyma, which might then adequately be submitted to illumination. Intraoperative PDT may not complicate the surgical procedure, which would be prolonged by no more than 20–30 min (based on our observations with adjuvant PDT performed during surgical resection of glioblastoma), and may also help the surgeon by providing fluorescence-guided imaging of the tumor at the beginning of the operation. Furthermore, even though ambient light in the operation room might induce photo-bleaching, this should not be a significant issue, given the lack of specificity and low energy of the ambient light which shall require a significantly longer illumination duration in order to induce significant cytotoxicity, compared to the red-light used during PDT [57]. We also believe that intraoperative adjuvant PDT may be of interest in terms of postoperative early recurrence, irrespective of the expected surgical margins. The 3-year and 5-year recurrence-free survival after R0-resection of HCC is still about 50% and 40% respectively, with no impact of surgical margin being superior to 10 mm [58] and indeed, many other studies showed that surgical margins do not influence the postoperative recurrence rates, overall survival, or recurrence pattern [59,60,61]. In addition, PDT will induce an anti-tumor immune response, which might further eliminate the possibility of tumor recurrence thereby giving long-lasting protection through the development of an immune memory, which will be discussed in the following section.

However, current clinical data regarding the use of PDT in HCC patients are scarce, consisting only of small patient groups with short-term follow-up. Furthermore, there has been no study that correlates the efficacy of PDT with the cause of HCC. Particularly, PDT has not been tested yet on in-vivo HCC models arising in the context of non-alcoholic steatohepatitis (NASH). Given the increasing number of HCC in NASH cirrhosis, especially in western countries, this is certainly a field where clinicians and researchers should come together to design relevant clinical trials in the future. From a theoretical point of view, one might expect a difference in the efficiency of the PDT in NASH-HCC as compared with viral-HCC, as NASH-HCC more frequently arises before end-stage fibrosis [62] and, on the other hand, tumors are often larger than in other etiology. From a surgical point of view, in operable patients, the results of surgical resection tend to be slightly better in NASH-HCC as compared with viral-HCC [63] and thus, justify the idea of combining intraoperative PDT and liver resection in those patients to further optimize the results. It should be noteworthy that such procedures are applicable only to candidates eligible for hepatectomy i.e., BCLC stage 0 or A patients with early-stage solitary non-cirrhotic tumor, with good liver performance and no radiological evidence of vascular invasion.

## 4. Photodynamic Therapy May Induce an Anti-Tumor Immunity

For most of the anti-neoplastic modalities, there is a change in the organization of the infiltrating immune cells in the tumor microenvironment, which can be crucial for the overall follow-up of the therapy. This change can be either pro or anti-tumoral, which thereby gives the tumor either resistance and cause tumor recurrence or induce a long-lasting anti-tumor immune response resulting in better overall patient survival, respectively. An ideal anti-cancer modality will not only destroy the tumor but also trigger the immune system to work against the neoplasm, either primary or malignant, by inducing Immunogenic Cell Death (ICD). As postulated by Kroemer et al., ICD is induced by cytotoxic therapies which induce calreticulin exposure, ATP secretion, and release of HMGB1 (High Mobility Group Box 1), HSP70 (Heat Shock Protein 70), and HSP90 (Heat Shock Protein 90) among others, which are preceded by either Endoplasmic Reticulum based stress, ROS production or autophagy [64]. These Damage Associated Molecular Patterns (DAMPs) induce activation and maturation of innate immunity and thereby induction of an immune response. Since ROS production is the *modus operandi* for PDT, the induction of such immune-stimulatory effects by PDT is quite obvious and relevant for involution and control of the neoplastic lesions. Various in-vivo studies have concluded that the efficacy of PDT is reduced in the absence of an active immune system, thereby highlighting that PDT has an immune-stimulatory impact which may have some clinical influence.

Like any host response to an external stimulus, PDT-induced immune response will rely on an intricate network of inflammatory cytokines, chemokines, transcription factors and release of DAMPs by the PDT treated tumor. After treating colon cancer with pyropheophorbide-a methyl ester-based PDT, two waves of transcription factor NF-κB (Nuclear Factor κB) activation were observed [65]. NF-κB regulates the expression of a wide range of genes responsible for the activation of inflammation and immune response. Along with transcription factor AP-1, NF-κB induces the expression of cytokines such as IL-1α, IL-6, IL-8, TNFα [66]. Thus, PDT activates pro-inflammatory mediators thereby generating an acute inflammatory response.

Studies involving various PS and cancer models have demonstrated the direct impact of PDT on immune components can be activating, suppressive, or lethal [67]. Garg et al. showed through an in-vitro set-up that when cancer cells are treated with reticulotropic PS, Hypericin based PDT, there is an exposure of calreticulin and HSP70, which then facilitate the tumor cell phagocytosis by Dendritic Cells (DCs), thereby highlighting the underlying mechanism of ICD by PDT [68,69,70].

This immune-modulatory effect of PDT in a BALB/cJ mice based in-vivo model was demonstrated by Korbelik and Cecic. Using Photofrin^®^ mediated PDT coupled with mycobacterium cell-wall extracts, they highlighted a decrease in the re-occurrence or a relapse of mammary sarcoma. This study underlines that the inflammatory response triggered by PDT can be augmented by an adjuvant, thereby giving a surge of anti-tumoral cytokines such as TNF-α, IL6 [71]. Later it was demonstrated that PDT mediated by another PS, 2-[1-hexyloxyethyl]-2-divinyl pyropheophorbide-a (HPPH), can induce a similar inflammatory response by local secretion of Macrophage Inflammatory Protein 2 (MIP-2) and E-selectin which causes an influx of neutrophils in the microenvironment capable of tumoral cytotoxicity, and thereby recruitment of other immune cells by secretion of cytokines and chemokines [72]. These cytokines can have other impacts, for example, decreased IL-10 secretion which further inhibits skin contact hypersensitivity [73]. This thereby highlights the role of cytokines and chemokines along with other secretory factors, in inducing inflammation and overall action-reaction scenarios for PDT.

Generally, PDT-induced tissue damage causes the infiltration of innate immune cells due to underlining oxidative stress, which leads to increased expression of Hypoxia Inducible Factor (HIF) [68]. Additionally, since this lowered level of oxygen typically resembles a site of wound or infection, HIF also causes secretion of other inflammatory cytokines and co-stimulatory factors to enhance the function of these infiltrating innate cells [68]. Being the first mediators of an immunologic response, they primarily include neutrophils, macrophages, natural killers, DCs, and mast cells. Zhang et al. demonstrated that DCs matured and activated by deuteporfin-mediated PDT on mice hepatomas could significantly decrease the tumor growth along with higher survival rates when compared with PDT alone [49]. Here, DCs, the professional antigen-presenting cells, engulfed the tumor-associated antigen released by the PDT, which thereby activated and presented effector T cells to induce an anti-tumor immune response finally. 

However, activation of adaptive immunity is the most important aspect, in order to impart a long-lasting tumor growth control. In light of that, Korbeliek et al. demonstrated an adoptive transfer of splenocytes from mice treated with Photofrin^®^ PDT against mammary sarcoma, which resulted in increased tumor regression post-PDT in the recipient SCID mice compared to the mice receiving just the PDT dose. This highlights that the presence of tumor-sensitized T cells in the spleen can have a significant impact on augmenting the impact of PDT [74]. Furthermore, Kabingu et al. demonstrated that after treatment of a sub-cutaneous mammary tumor with Photofrin^®^ PDT in BALB/cJ mice resulted in tumor decrease of primary as well as secondary tumor sites in the lungs by increased infiltration by CD8+ T cells [75]. These results highlight that besides the direct cytotoxicity, PDT induces an anti-tumor vaccine, by the generation of memory CD8+ T cells.

## 5. PDT and Immune Response in HCC

Due to viral infection and cirrhosis, a majority of patients suffer from chronic inflammation in HCC. In the tumor microenvironment, there are a high prevalence of immuno-suppressive regulatory T cells (Tregs) and Myeloid-derived suppressor cells (MDSCs) (monocytes, macrophages, and DCs), along with an increased expression of immune checkpoint regulators [76]. Due to these immune suppressive populations, the tumor-infiltrating CD8+ T cell population gets exhausted and their capacity to present tumor-associated antigen is impaired, which further leads to tumor progression and poor prognosis [76,77,78]. All this develops a network of cytokines, chemokines, and other factors resulting in an intricate microenvironment. With the recent development in immune checkpoint inhibitor-based immunotherapy, the influence of the suppressive population in tumors has decreased. The two key targets are Programmed Death Ligand 1 (PD-L1) and Cytotoxic T Lymphocyte Antigen 4 (CTLA4). When these inhibitory signals bind to their receptors on T cells (CD8+ and/or CD4+), it reduces their proliferation. At the same time, they also reduce Treg apoptosis and contribute to their inhibitory function [76]. These signals are often overexpressed in the tumor microenvironment, thus contributing to the immune escape mechanism. Blockage of these signals, by using anti-PD-L1 and anti-CTLA4 antibodies, has given improved results in the clinic for a wide range of solid tumors, alone or in combination with existing chemo or radiotherapy. Regarding HCC management, the recent report on the combination of *atezolizumab* (a PDL-1 inhibitor) with *bevacizumab* (a VEGF inhibitor) [79], showing its superiority against *sorafenib*, has been a major step towards the use of immunotherapy as the first-line systemic treatment of advanced HCC. Thus, combining such immunotherapy strategies with PDT could be a relevant proposition. Immune checkpoint blockade-based immunotherapy along with PDT could increase infiltration of tumor-specific effector T cells and decrease secretion of TGFβ, an immunosuppressive cytokine secreted by Tregs which have an autocrine role [66], might lead to lower tumor recurrence and higher patient survival rate. However, a lot of research is still needed in both immunotherapy and PDT fields, especially for PDT dose and treatment standardization, along with guidelines from respective associations, before we could initiate a combinatorial approach for PDT and immunotherapy.

The basic rationale for HCC treatment is the targeting of the primary tumor site along with the suppression of pro-tumor factors. The current treatment regimens only target one of the aspects of the rationale, while the persistence of the immune-suppressive microenvironment remains a hurdle. PDT causes tumor destruction, which results in a tissue injury and therefore release of tumor antigen. This initiates a host–tumor reaction, which results in infiltration of Tumor-Infiltrating Lymphocytes (TILs) to induce an anti-tumoral immune response and can be combined with immunotherapy to augment its impact. Hence, PDT will not only target the HCC, but it will also transform its microenvironment from a pro-tumoral to anti-tumoral. This was proven in HCC based in-vitro study where it was reported that Pheophorbide-mediated PDT induces ICD by triggering phagocytosis of cancer cells by macrophages. These macrophages will thereby process and present the tumor-associated antigens with HLA proteins and HSP70 resulting in an antigen-specific T cell stimulation in the host [50]. As described previously, during ICD, the anti-tumoral modality shall induce the release of immunoreactive molecules such as tumor-associated antigens, along with DAMP signals (such as HSP70, HSP90, ATP, HMGB1). This shall attract and activate innate immune cells such as macrophages and DCs to engulf these signal molecules, and thereby act as antigen-presenting cells. For instance, when macrophages phagocytose dying cancer cells, they evolve into activated anti-tumoral M1 type cells, thereby helping in creating and maintaining an anti-tumoral microenvironment [64,67].

## 6. Conclusions

Recently, PDT has received growing attention in the international community, which is evident from the rising number of publications, but even though the therapy has seen a lot of advances in almost all of the fields, a lot of work still needs to be carried out especially with the combinatorial approach of PDT. PDT-induced anti-tumor effects include direct tumor cytotoxicity, tumor-infiltrating immune cells, innate immune cell recruitment, and vasculature shut down. Since HCC occurs in the background of chronic inflammation with a complex microenvironment, the role of PDT becomes of interest as it has shown the potential to transform an immuno-suppressive environment into an anti-tumoral one.

Through this review, we would like to suggest that PDT should be regarded as an intraoperative adjuvant procedure during HCC resection (Figure 1). Once the PS is administered, followed by an optimal drug-to-light interval, the fluorescence generated by PS excitation shall provide a visual aid to the surgeon to detect infra-clinical nodules and to guide liver resection. At the end of the surgical procedure, the cavity shall be illuminated by the light of appropriate power and wavelength, using optimal optic-fiber and laser-based illumination devices. Intraoperative PDT may not only kill the undetected residual tumor but also activate a possible immune response. This immune response can be even more effective when combined with the administration of an immune checkpoint blockade-based immunotherapy.

However, before we proceed further to couple PDT with immunotherapy or any other existing therapeutic options for HCC, we need to standardize PDT protocol for HCC in a clinical set-up, which shall include optimal PS dose, optimal drug-to-light interval, optimal light dose, and fractionation protocol. We would also like to accentuate the necessity of developing illumination devices specifically designed to be used during open or laparoscopic hepatectomy, PS with enhanced tumoral selectivity, along with more novel therapeutic strategies for the treatment of HCC. The application of improved protocols for adoptive therapy in combination with PDT, which could be facilitated by the chemotactic factors secreted by the treated tissue, may also yield higher efficacy for HCC treatment in the future.

BCLC: Barcelona Clinic Liver Cancer staging system; HCC: Hepatocellular Carcinoma; PS: Photosensitizer; PDT: Photodynamic Therapy; AASLD American Association for the Study of Liver Diseases; EASL: European Association for the Study of the Liver.

## Figures and Tables

**Figure 1 cancers-13-05176-f001:**
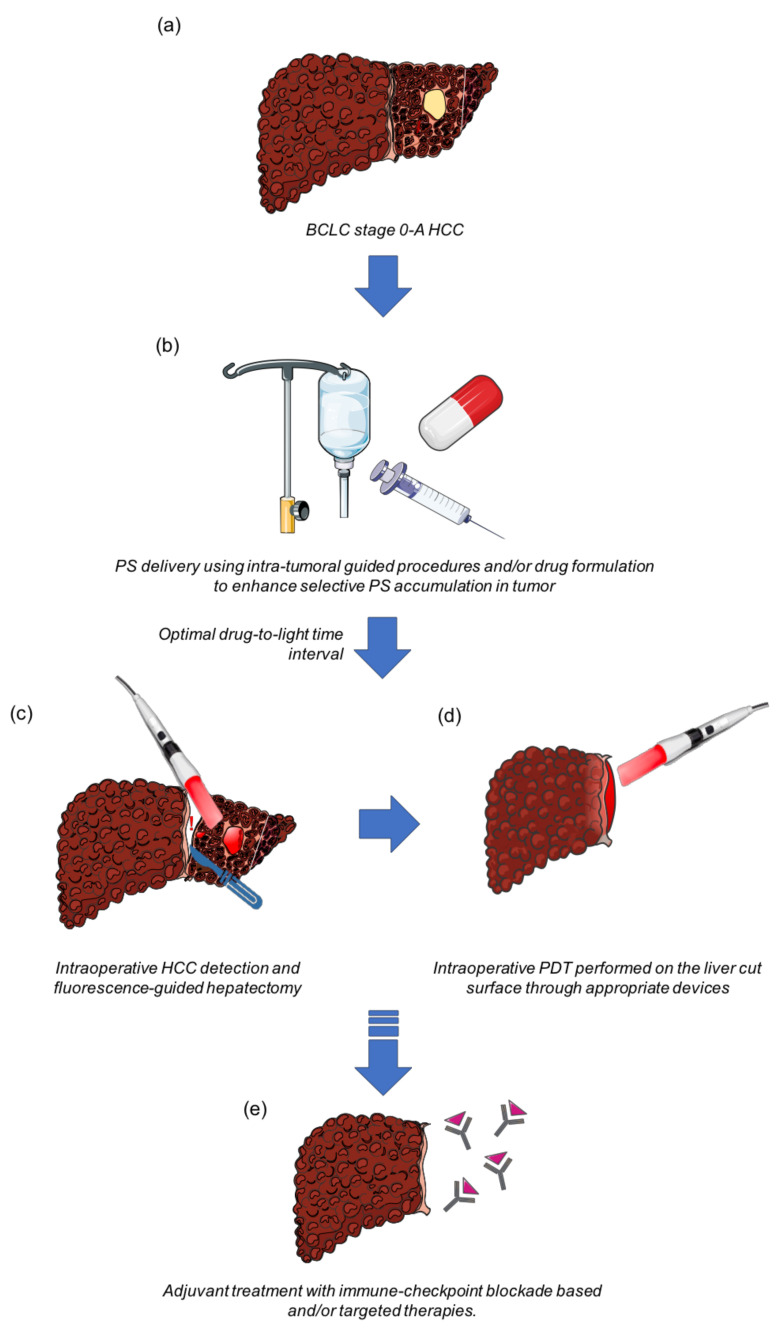
Proposed strategy for using intraoperative PDT as an adjuvant treatment in liver resection for HCC. (**a**) Patient’s eligibility criteria for liver resection should be those recommended by the AASLD-EASL international guidelines; (**b**) prior to the hepatectomy, a PS (or its biological precursor) shall be administered using either loco-regional or systemic route to enhance the tumoral accumulation of the PS; (**c**) after an optimal drug-to-light interval, the PS generated fluorescence shall be used to identify previously undetected tumors, and to provide a visual aid for the surgeon during hepatectomy; (**d**) thereafter, at the end of hepatectomy, PDT protocol specific to the PS wavelength and dose shall be initiated onto the tumoral bed to remove any residual tumors; (**e**) the PDT-induced anti-tumoral immune response might be even more efficient if the patient could subsequently be offered an immune checkpoint blockade-based immunotherapy, thus increasing disease-free survival.

**Table 1 cancers-13-05176-t001:** List of key clinical and pre-clinical studies using Photodynamic Therapy for the treatment of various cancers. PS: Photosensitizer; VEGF: Vascular Endothelial Growth Factor; PDT: Photodynamic Therapy; 5-ALA: 5-Aminolevulinic Acid; PpIX: Protoporphyrin IX.

Photosensitizer	Cancer	*n*	Conclusion	Ref
Hematoporphyrin Derivatives	Carcinomas of the breast, colon, prostate, squamous cell, basal cell, and endometrium; malignant melanoma; mycosis fungoides; chondrosarcoma; and angiosarcoma	24	Highly pigmented and larger sub-cutaneous tumors require a stronger dose (5 mg/kg of PS) than non-pigmented and superficial tumors (2.5 mg/kg of PS); skin damage reduced by reducing illumination dose; maximum tumor necrosis observed till 2 cm.	[16]
Hypericin	Bladder carcinoma	(pre-clinical)	PDT in combination with anti-VEGF (bevacizumab) increases tumor responsiveness and reduced VEGF expression along with downregulation of other angiogenic proteins.	[31]
Photofrin^®^	Breast cancer with chest wall progression	14	A low dose of Photofrin^®^ (0.8 mg/kg) mediated PDT induced tumor necrosis with lesions >2 cm in thickness; initial regression of untreated lesion was observed; Wound care-related difficulties were observed.	[33]
Photofrin^®^, Photogem, Photosan-3, or Temoporfin	Unresectable cholangiocarcinoma	402 (meta-analysis)	PDT with biliary stenting could significantly improve patient survival period.	[34]
Photosan^®^, Photofrin^®^, or Foscan^®^	Unresectable extrahepatic cholangiocarcinoma	96	Combination of PDT with systemic chemotherapy showed significantly longer overall survival than chemotherapy alone with higher median survival than control groups. The therapy was well tolerated.	[35]
Photofrin II	Metastatic breast cancer	37	PDT yields the best results in patients with asymptomatic lesions; reductions in Photofrin^®^ dose with reciprocal increases in light dose did not impair treatment efficacy.	[36]
5-ALA	Ovarian carcinoma metastases	29	Laparoscopic fluorescence detection of PpIX after intraperitoneal application of 5-ALA; histological assessment of the biopsy specimens proved that strong red fluorescence had a sensitivity of 92% for detecting tumor tissue on specimens.	[37]
5-ALA	Glioblastoma	10	Intraoperative PDT following PpIX fluorescence-guided maximal resection and adjuvant therapy resulted in an increased overall survival rate; no adverse effects were observed.	[38]
5-ALA	Malignant Pleural Mesothelioma	20	Combination of 5-ALA PDT with thoracoscopy followed by Anti-PD-1 *Nivolumab* immunotherapy for a maximum of 2 years; Currently ongoing.	NCT04400539

**Table 2 cancers-13-05176-t002:** List of various photosensitizers tested on Hepatocellular Carcinoma, hepatoma or healthy liver for Photodynamic Diagnosis, or Photodynamic Therapy or both. PS: Photosensitizer; PDT: Photodynamic Therapy; HCC: Hepatocellular Carcinoma; mTHPC: meta-tetra (hydroxyphenyl) chlorin; mTHPBC: 5,10,15,20-tetrakis (m-hydroxyphenyl) bacteriochlorin; 5-ALA: 5-Aminolevulinic Acid; ICG: Indocyanine Green.

Photosensitizer	Study Set-Up	Conclusion	Ref
mTHPC	*In-Vivo*	High tumoral accumulation of the PS was observed in rat liver metastases with respect to the normal liver; extensive tumor tissue damage upon illumination; mild and transient damage to normal tissue was observed.	[41]
Photofrin, mTHPC, and mTHPBC	*In-Vivo*	Upon illumination, near-infrared PS mTHPBC showed significantly larger necrotic areas than the others in normal rat livers; highlight the advantage of near-infrared PS activation for pigmented tissues like the liver.	[43]
New nanocarrier containing IR780	*In-Vitro and in-vivo*	IR780 and Paclitaxel (chemotherapeutic drug) loaded nanocarriers exhibited synergistic effect by inducing cancer cell apoptosis and cell cycle arrest at the G2/M phase for HCC; the combined treatment inhibited the in-vivo tumor growth and the tumor angiogenesis.	[44]
Chlorin e6 containing gold nanoparticles	*In-Vitro and in-vivo*	PDT coupled with hypoxia-induced chemotherapy showed a synergistic anti-HCC effect.	[45]
ICG-loaded lactosomes	*In-Vitro and in-vivo*	ICG-lactosome PDT treated HCC cells have higher cytotoxicity than ICG PDT; ICG-lactosome had higher fluorescence of tumor areas than ICG alone, along with anti-neoplastic effects on these malignant implants.	[46]
5-ALA	*In-Vitro and in-vivo*	In-Vitro and in-vivo PpIX fluorescence was detected in tumors; red fluorescence was detected in HCC patient samples who were orally administered with 5-ALA before resection.	[47]
5-ALA	*In-Vivo*	Higher PpIX fluorescence intensity was detected in HCC than in non-tumoral tissues in Male Fisher-344 rats; PDT induced necrosis in tumoral tissue; no necrosis was evident in non-tumoral tissue.	[48]
Deuteporfin	*In-Vivo*	PDT can inhibit mouse hepatoma growth and induce an anti-tumor immune response.	[49]
Pheophorbide-a	*In-Vitro*	PDT caused tumoral cytotoxicity of HCC cell lines by induction of apoptosis; PDT-induced immunogenicity triggered phagocytic capture of HCC cell lines by human macrophages.	[50]

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
