# Peer review of "Could Photodynamic Therapy Be a Promising Therapeutic Modality in Hepatocellular Carcinoma Patients? A Critical Review of Experimental and Clinical Studies"

_cancers, 2021, doi:10.3390/cancers13205176_

Round 1

Reviewer 1 Report

Very well written manuscript with very interesting concepts for HCC-treatment.

  1. Are there any differences in the effectiveness of PDT dependent on the cause of cirrhosis (ASH/NASH vs. viral hepatitis). Please highlight this matter
  2. Is there any proposal for a standardized adjuvant protocol consisting of PDT + immuncheckpoint inhibitor after hepatectomy for HCC. What is the experience when comparing adjuvant procedure and intraoperative PDT? Is there any superiority when PDT is being performed during surgical procedure? What do the authors think about it?
  3. Do the authors recommend also a PDT also in cases of R0-resection with wide margins (for example >1cm)?

Author Response

We have also answered all comments of the two reviewers (the full copy of the questions and comments are in black font and the answers are in red font) and have amended the manuscript when requested (changes are marked up using the track changes function of MS Word). 

Reviewer 2 Report

The review of Kumar et al. aims to present a critical review of experimental and clinical studies concerning the potential of photodynamic therapy (PDT) as a therapeutic mean in patients with hepatocellular carcinoma (HCC). The topic is of clinical interest and fits the scope of the journal. Indeed, PDT therapy might be an important additional therapeutic venue for high-grade HCC that currently lacks convincing curative options.

The review could be a valuable addition to the journal however, the manuscript needs some improvement before it attains the level of publication.

Major points:

  1. The review suffers from a lack of clear structure: in several chapters authors alternate back and forth between in vitro experiments, animal models and human trials. It makes reading the review somewhat confusing. It would be helpful to have a more structured presentation.
  2. It would greatly enhance the quality of the review if authors could present a summary table about the studies mentioned in the manuscript with the positive and negative sides highlighted.
  3. It is also recommended to present a table with the different photosensitizers (PS), noting the ones that might be promising in the treatment of HCC.
  4. Authors should describe more precisely how PDT acts, most importantly note that the formation of H2O2 and OH- are follow-up reactions and not directly formed by the interaction between the PS and light.
  5. Page 4, Line 364-68: “Hence, PDT will not only target the HCC, but it will also transform its microenvironment from a pro-tumoral to anti-tumoral. This was proven in HCC based in-vitro study where it was reported that Pheophorbide based PDT induces immunogenic cell death by triggering phagocytosis by macrophages [59].”

It is not clear which cells are phagocytosed by the macrophages and why would it constitute an anti-tumoral environment.

Minor points:

  1. The “Abstract” part should use present tense and not past tense
  2. In general, the manuscript should be revised for grammatical errors. It contains a lot of sentences with mismatched noun in plural and verb in singular forms and vice versa.
  3. Page 2, Line 43: It would be helpful to state the percentage of stage 1-4 of HCC patients at time of diagnosis.
  4. Page 2, Line 48: Authors cite an example for locoregional therapies (chemoembolization) but not for the targeted therapies.
  5. Page 2, Line 51: “HCC recurrence rates are quite high” please provide more precise data and references.
  6. Page 2, Line 65: “Reactive Oxygen Species” please change to reactive oxygen species (without capital letters).
  7. Page 6; Lines 274, 276 and 286: spell out HMGB1, HSP70, HSP90 and DAMP and NF-κB
  8. Page 7, Line 308: Hypoxia-Inducible Factor (instead of “Hypoxia Inducing Factor)
  9. Page 8, Line 361: this sentence does not make any sense, please rewrite:“Since PDT, causes a tumor insult, which results in a tissue injury and release of tumor antigen”.

Author Response

(The authors gave the same response as above.)

Round 2

Reviewer 1 Report

All my points/questiones have been answered very well. I was glad to receive such an excellent revision.